# Hydro–Mechanical Behaviour of a Rainfall-Induced Landslide by Instrumental Monitoring: Landslide–Rainfall Threshold of the Western Black Sea Bartin Region of Türkiye

**Taha Taskiran [1,\*], Serdar Alli [2,3] and Yuksel Yilmaz [4]**

1 Department of Civil Engineering, Ankara Yildirim Beyazit University, Ankara 06010, Türkiye
2 Department of Civil Engineering, Bartin University, Bartin 74100, Türkiye; salli@bartin.edu.tr
3 Graduate School of Natural and Applied Sciences, Gazi University, Ankara 06500, Türkiye
4 Department of Civil Engineering, Gazi University, Ankara 06570, Türkiye; yyuksel@gazi.edu.tr
\* Correspondence: ttaskiran@ybu.edu.tr

**Abstract:** Bartin City is located in the Western Black Sea Region of Türkiye, where rainfall-induced landslides are more frequently observed. Although it is known that many landslides are induced by rainfall, there is limited knowledge regarding how rainfall triggers these landslides in the city. To clarify the triggering mechanisms of rainfall-induced landslides, a detailed field monitoring program was performed on a chosen area to represent landslides in Bartin. The instrumentation included the measurements of site suction, volumetric water content, groundwater level, and rainfall amount over a period of two years. Various stability analyses were performed regarding pore pressures after both transient flow infiltration analysis and site-measured suction values. The rainfall intensity–duration thresholds were obtained for both dry and wet periods as a result of the numerical analyses performed by means of parameters obtained from field monitoring. The results show that the wet period conditions create more critical conditions before failure compared to the dry period conditions, so landslides occur more easily in wet periods. According to the landslide–rainfall threshold relations, landslide-risk limits are reached if the rainfall intensity is over 10 mm/h for the dry periods and lasts between 0.85 h and 17 h depending on the saturated hydraulic conductivity of the soil. When the rainfall intensities are less than 10 mm/h, longer rainfall durations are needed for a landslide to occur. For the wet periods, landslide-risk situations are encountered if the rainfall intensity over 1 mm/h continues for 0.36 h–3.67 h, depending on the saturated hydraulic conductivities.

**Keywords:** landslide; instrumental monitoring; rainfall threshold; matric suction; early warning; data collection & analysis

## 1. Introduction

Rainfall-triggered landslides are one of the natural disasters that cause serious loss of life and property [1–6]. Almost every year, people living in many countries of the world are affected dramatically due to rainfall-induced landslides. Governments have to spend millions of dollars from their budgets to repair the damage caused by landslides. Due to global warming and climate changes, the expected increase in the number of landslides triggered by rainfall in the coming decades further raises concerns on this issue [7,8]. Clarifying the triggering mechanisms of rainfall-induced landslides can contribute to a reliable estimation of the occurrence times of landslides and early warning systems.

Rainfall intensity–duration (I-D) thresholds play a significant role in the prediction and early warning of landslide events. I-D thresholds for landslides are revealed by examining the rainfall resulting in landslides [9–13]. The threshold was obtained by determining the lower limit of the rainfall duration and rainfall intensity components that cause landslides plotted in logarithmic, semi-logarithmic or cartesian coordinates [14]. Global, regional and local studies on I-D thresholds have been carried out in various studies

in the literature [9,11,14–19]. The thresholds can be established using empirical or physical methods. Empirical rainfall thresholds are defined by statistically studying past rainfall events that led to landslides. While the rainfall thresholds for landslides can be helpful in assessing landslide susceptibility and risk, they have certain limitations that can lead to false and missed alarms [13,19–22]. Empirical landslide–rainfall threshold relationships can significantly affect the reliable prediction of the occurrence times of landslides since they do not usually consider non–triggering rainfall, antecedent conditions, landslide mechanism, and soil geotechnical properties [22–26]. A well-defined rainfall threshold model should aim to achieve both high accuracy in detecting actual landslide events and minimize false alarms [20,21]. Physically based methods use infiltration models (e.g., steady-state, transient-state) to relate rainfall events to slope stability analysis. Providing a theoretical framework for the slope failure process due to rainfall infiltration, physically based methods attempt to predict temporal and spatial distributions of rainfall-induced landslides through Geographic Information Systems (GIS) [27–29]. Despite its usefulness for future research, the method also has certain limitations. Comprehensive spatial data on hydrological, geological, morphological, and soil properties are necessary to simulate landslide initiation for physical methods accurately. Accurately gathering this information over extensive areas is challenging, and it is seldom accessible. The current study can provide a database with extensive instrumental monitoring work.

On the other hand, saturated soil mechanics methods are also frequently used in landslide analysis. However, these methods do not consider the unsaturated strength and unsaturated hydraulic properties of the soils. Unsaturated soils contain water and air in their pore spaces. Unlike saturated soils, where all the voids are filled with water, unsaturated soils have varying degrees of water saturation. Factors such as matric suction, moisture content and unsaturated permeability play significant roles in the behavior of unsaturated soils. When analyzing the behavior of unsaturated soils, relying solely on saturated soil mechanics can be insufficient [23,30–32].

Depending on the precipitation and evaporation cycle on the earth, there are various changes in water content, matric suction and underground water level in the soil. Depending on these variations, the shear strength of the soil changes. Studies on the stability of slopes showed that the infiltration of rainfall into unsaturated soil slopes significantly affects the stability of the slopes [33–40]. Soils that are not initially saturated have a certain matric suction before the rainfall. This matric suction has an increasing effect on the shear strength of the soil. With the infiltration of rainfall into the unsaturated soil, the water content and saturation of the soil start to increase, and the matric suction tends to decrease. Instability can occur when the matric suction at the critical slip surface of the slope decrease or when the matric suction is completely lost, and positive-pore water pressures develop. The development of this behavior is a very complex issue depending on various parameters such as rainfall intensity, rainfall duration, slope geometry, hydro–mechanical properties and initial boundary conditions (groundwater level, water content, degree of saturation, matric suction, etc.) [41–43]. The clarification of this mechanism for various geographical regions where rainfall-induced landslides are observed will make significant contributions to the landslide literature.

In this study, a representative landslide area was determined in order to elucidate the hydro–mechanical behavior of landslides in Bartin. In order to investigate the triggering mechanism of rainfall-induced landslides, instrumental monitoring studies that lasted for two years, laboratory tests, and numerical analyses to establish accurate parameters and, thus, to establish the triggering mechanism more realistically were performed. The variations in matric suction, volumetric water content, groundwater level, and rainfall amount were measured with sensors in the landslide area. The geotechnical properties of the ground were determined by laboratory tests. Using data obtained from in-situ monitoring and laboratory tests, stability analyses were performed. The landslide was investigated by means of additional parametric analyses performed based on transient flow analyses using the rainfall data and subsequent stability analyses. The rainfall intensity–

duration thresholds were obtained for both dry and wet periods separately as a result of the numerical analyses performed by considering the site-specific measured seasonal matric suctions and groundwater levels.

## 2. Site Description

Bartin is located in the Western Black Sea Region of Türkiye (Figure 1). Rainfall-induced landslides are quite frequently observed in the city. Topography, climate and ground profile provides a suitable environment for rainfall-induced landslides. In the studies conducted in the past, it has been reported that sudden and multiple landslides occurred in the city, especially after the rainfall events in 1985, 1998 and 2016. The landslides in 1985 occurred in April as a result of snowfall in the winter season and the subsequent rains. The landslides in 1998 developed as a result of continuous rain on 20–21 May. The reason for the landslides which occurred in 2016 was heavy rain on 12–13 August. Some singular landslide cases were also observed at various times in other years. According to GIS-based studies carried out for Bartin by the Türkiye Disaster Affairs General Directorate, landslides that occurred in the city were concentrated in the Cretaceous flysch deposits, which are defined as the Ulus formation. According to Reference [44], the Ulus district of Bartin, which gave its name to the Ulus formation, is the district where the most landslide cases occur in Türkiye. As a result of landslides in the past, significant damage to the infrastructure and superstructure occurred, and many agricultural and forest areas became unusable.

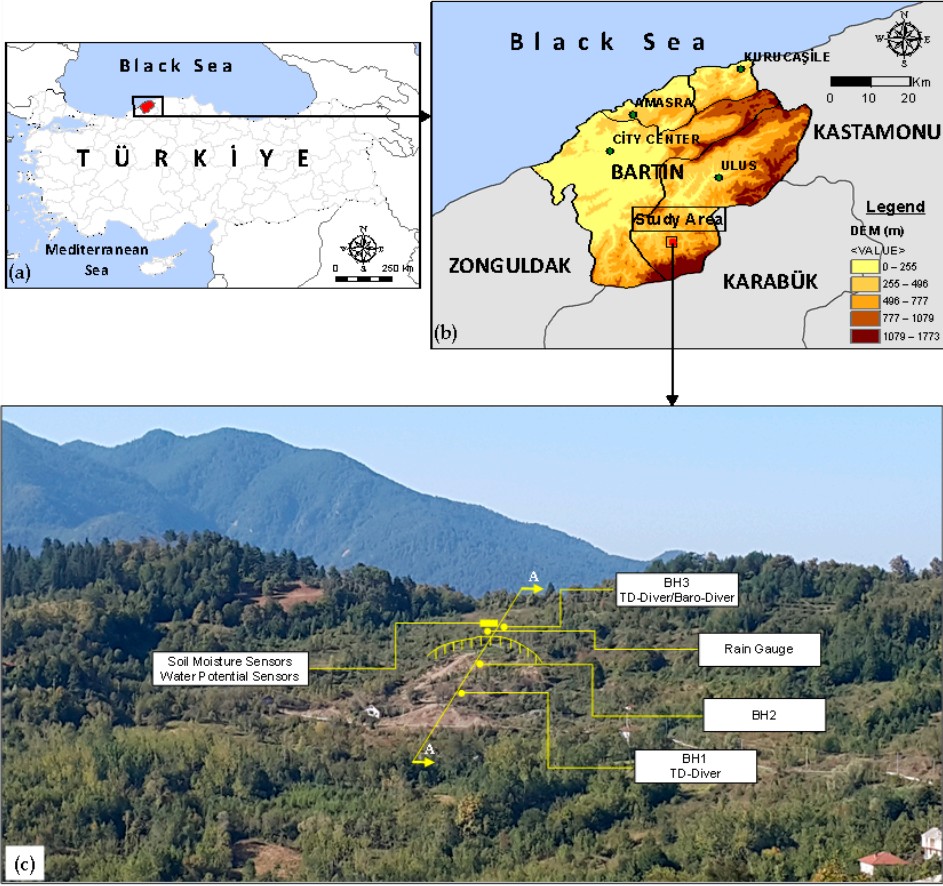

**Figure 1.** Location and general view of the Hisar landslide area (**a**) Türkiye, (**b**) Bartin, (**c**) Hisar landslide area.

In order to contribute to the understanding of the hydro–mechanical behavior, a representative landslide area was chosen in Hisar village of the Ulus district of Bartin. Hisar village is 42 km from the Bartin city center and 24 km from the Ulus district center. The

location and general view of the study area are given in Figure 1. The average slope gradient is 18°. In the location of the examined landslide, some tension cracks, undulations and shear failure surfaces were observed on the topography. The failed mass is approximately 70–75 m wide and 100–110 m long, and it was moved down by translational-type failure. From the observations made in the field, it is estimated that the depth of the slip surface is between 4.5 and 5.0 m. There is no house nearby the landslide area, only a single-story abandoned house located at the foot of the landslide was damaged at various scales due to the landslide.

## 3. Geotechnical Investigations

Subsurface ground investigation was performed by boreholes drilled to a depth of 20 m at the foot (BH1), 19.50 m in the main body (BH2) and 19.50 m in the crown (BH3) (Figures 1 and 2). Disturbed (D) and undisturbed (UD) soil samples and rock core samples were obtained for laboratory tests. Two piezometers were installed in the boreholes of BH1 and BH3 to examine groundwater level variations. UD soil samples were taken in accordance with ASTM D1587 [45] with Shelby tubes at sufficient numbers and depths to define the soil profile. The wireline drilling method using a double tube core barrel was used in rock formations.

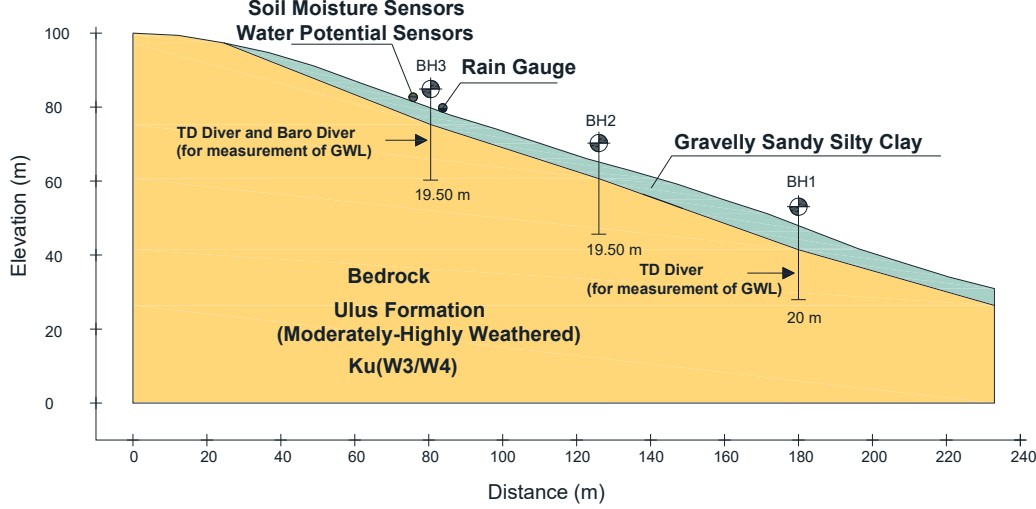

**Figure 2.** Hisar landslide area ground profile A–A section.

According to the subsurface investigation, the geotechnical profile of the Hisar landslide area is composed of moderately- to highly-weathered levels of the Ulus formation in the lower part and completely decomposed levels in the surficial part (Figure 2). The completely decomposed level is represented by a gravelly–sandy–silty–clayey soil unit with a yellowish-brown color, varying from medium to hard consistency. The unit is of moist, low plasticity, mostly siltstone/claystone origin, rarely sandstone origin. The lithologic unit of the bedrock in the landslide area is moderately to highly weathered levels of the Ulus formation. The Ulus formation represents a flysch sequence which consists of the alternation of siltstone and claystone interbedded with sandstone. The claystone is yellowish-brown, weak and highly weathered. The siltstone is green in color, yellowish-brown, weak and moderately to highly weathered. The sandstone is gray in color, medium strong and slightly weathered. The claystones and siltstones form layers of the same order, but bedding discontinuities are more common, depending on the fissility of the sediments.

## 4. Laboratory Tests

Sieve analysis–hydrometer [46], liquid limit–plastic limit [47], falling head permeability [48], consolidated undrained (CU) triaxial [49], pressure plate soil water characteristic curve (SWCC) tests [50] were carried out in the laboratory to determine geotechnical prop-

erties of soil samples taken from the field. Uniaxial compression tests [51] were performed on rock core samples.

When the grain size distributions are examined in Figure 3, it can be observed that they presented scattering. Grain size ranges: 2.3–34.0% for the gravel; 8.4–23.8% for sand; 20.9–38.4% for silt; and 18.4–54.7% for clay.

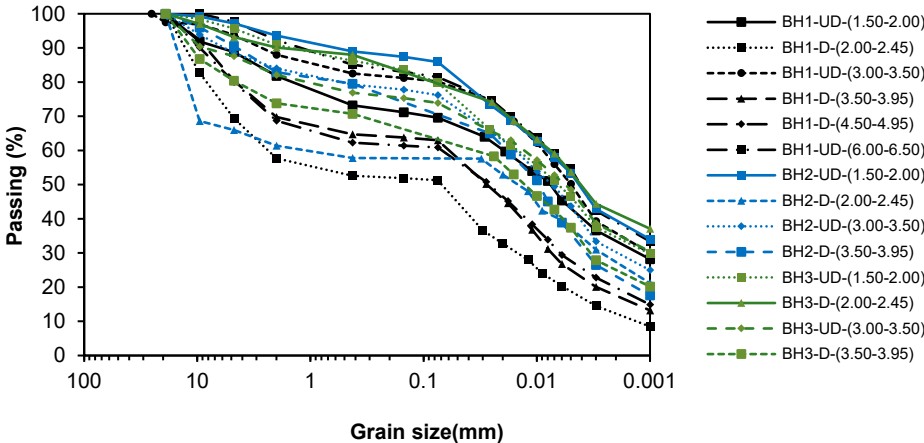

**Figure 3.** Grain size distributions of the soils.

The ranges of Atterberg limits are given in Table 1. The distribution of the soils in the plasticity chart is presented in Figure 4. As shown in Table 1 and Figure 4, most soils were classified as CL and distributed close to the A line in the plasticity chart.

**Table 1.** Atterberg limits and soil classes of Hisar landslide area.

| Borehole No | Depth (m) | Liquid Limit (%) | Plastic Limit (%) | Plasticity Index (%) | Soil Classification (USCS) |
|---|---|---|---|---|---|
| BH1 | 1.50–6.50 | 36–42 | 22–25 | 13–18 | CL |
| BH2 | 1.50–3.95 | 33–41 | 19–26 | 13–15 | CL, ML |
| BH3 | 1.50–3.95 | 40–54 | 24–29 | 15–30 | CL, CH, ML |

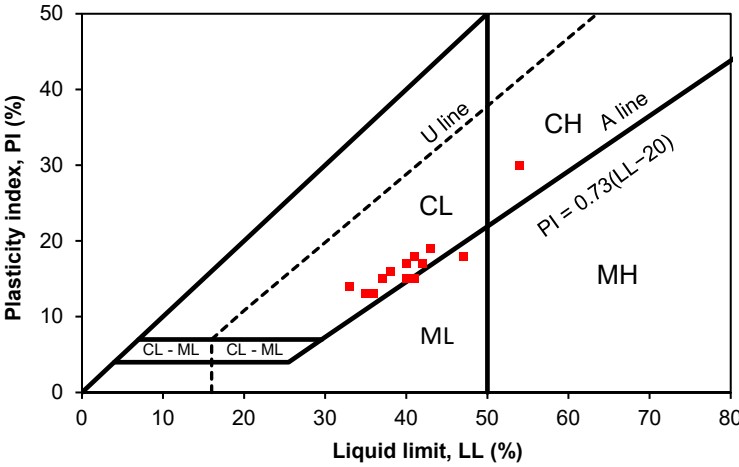

**Figure 4.** Representation of the soils in plasticity chart.

The saturated hydraulic conductivity coefficients of the soils ($k_{sat}$) were determined in the laboratory using the falling head permeability method in a rigid-walled permeability test setup on the undisturbed soil samples taken directly from landslide areas with permeability molds. Saturated hydraulic conductivity coefficients of the soils were calculated as an average of $1.76 \times 10^{-7}$ m/s. The effective friction angle (ø′) and effective cohesion (c′)

of the soils were obtained from the consolidated undrained (CU) triaxial tests. Since the undisturbed soil samples taken from the Shelby tubes contained the largest particle size, bigger than 1/6 of the specimen diameter, it was not possible to prepare a sufficient number of undisturbed test samples. Test specimens were prepared by compacting at in-situ dry density and in-situ water content. The graphs obtained as a result of the consolidated undrained (CU) triaxial tests are presented in Figure 5. The effective friction angle and the effective cohesion were 21° and 8 kPa, respectively.

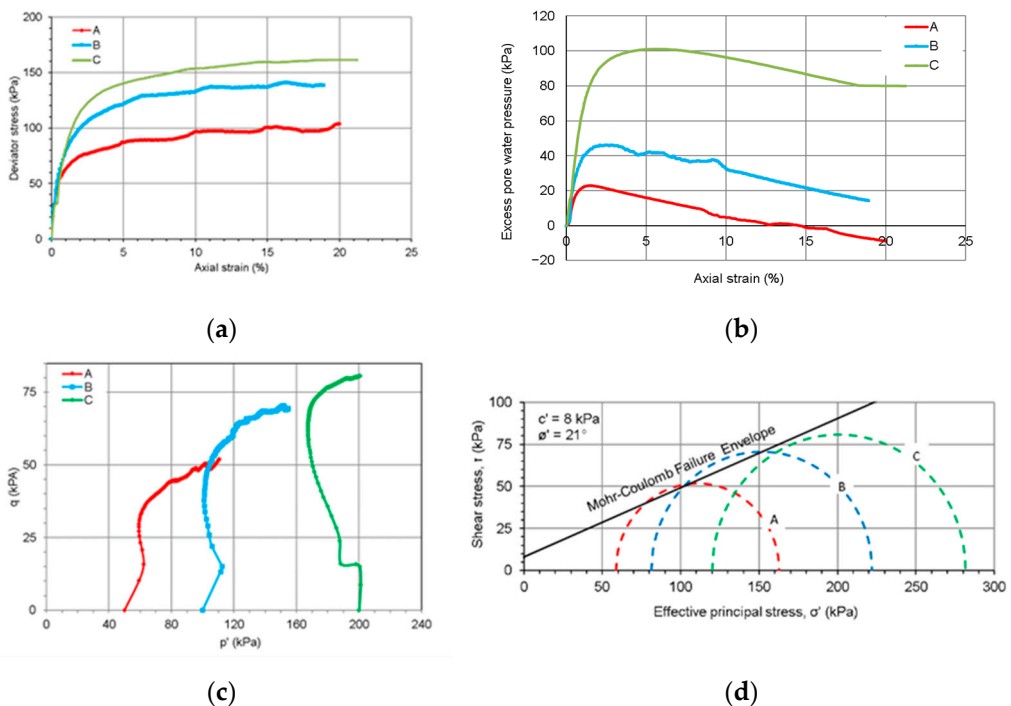

**Figure 5.** Consolidated undrained triaxial test graphs. (**a**) Deviator stress–axial strain, (**b**) Excess pore water pressure–axial strain, (**c**) Stress path, (**d**) Mohr circles.

SWCC, which provides the relationship between matric suction and water content, is an important parameter in the behavior of unsaturated soils. Studies show that there is a close relationship between the soil water characteristic curve and the engineering properties of unsaturated soils [32,52]. Depending on the precipitation and evaporation cycle, various changes occur in the water content, pore pressure, permeability, and shear strength in unsaturated soils. Previous studies showed that these changes can be calculated based on the parameters obtained from the shape of the SWCC. SWCC tests were performed using a pressure plate, according to ASTM D6836 [50], for drying. The tests were carried out on undisturbed soil specimens. The following formula proposed by Van Genuchten [53] was used to plot the best-fit curve of the SWCC, which provides the relationship between matric suction and volumetric water content.

$$\Theta = \Theta_r + \frac{(\Theta_s - \Theta_r)}{[1 + (\alpha(u_a - u_w)^n]^m} \tag{1}$$

where $\Theta$ is the volumetric water content; $\Theta_r$ is the residual volumetric water content; $\Theta_s$ is the saturated volumetric water content; $\alpha$, n and m are the fitting parameters; $(u_a - u_w)$ is the matric suction.

Van Genuchten curves fitted to the pressure plate test measurement points are given in Figure 6. Fitting parameters, $\Theta_s$ and coefficient of determination $\left(R^2\right)$ values were determined as shown in Table 2. As observed in Figure 6, SWCCs have different shapes even in the same landslide area. In previous studies, it has been stated that the shape of the SWCC can vary depending on many soil properties such as pore size distribution, grain size

distribution, density, organic matter content, clay content and mineralogy. The SWCCs of the soils obtained from the landslide area had different shapes due to the heterogeneity of the soils. It is compatible with the literature in this sense. The average of the fitting curves was used for the representation of the SWCC in subsequent seepage and stability analyses.

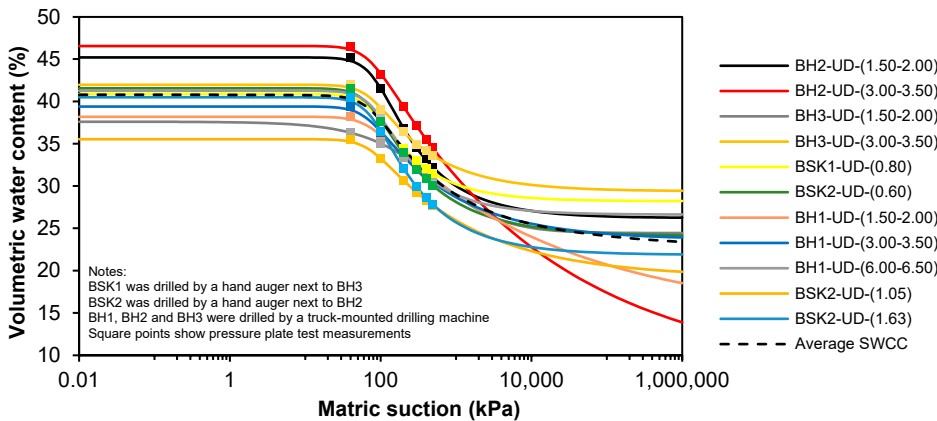

**Figure 6.** Van Genuchten curve fittings to test points of soil water characteristic curves.

**Table 2.** Fitting parameters, $\Theta_s$ and coefficient of determination $\left(R^2\right)$.

|  | $\alpha$ | n | m | $\Theta_s$ (%) | $R^2$ |
|---|---|---|---|---|---|
| BH1-UD-(1.50–2.00) | 0.016 | 2.997 | 0.059 | 38.2 | 0.996 |
| BH1-UD-(3.00–3.50) | 0.015 | 3.0 | 0.141 | 39.4 | 0.996 |
| BH1-UD-(6.00–6.50) | 0.012 | 2.995 | 0.244 | 41.3 | 0.998 |
| BH2-UD-(1.50–2.00) | 0.012 | 3.0 | 0.217 | 45.2 | 0.998 |
| BH2-UD-(3.00–3.50) | 0.014 | 2.982 | 0.063 | 46.6 | 0.998 |
| BH3-UD-(1.50–2.00) | 0.001 | 0.844 | 1.370 | 37.6 | 0.995 |
| BH3-UD-(3.00–3.50) | 0.014 | 2.956 | 0.194 | 42.0 | 0.997 |
| BSK1-UD-(0.80) | 0.013 | 3.0 | 0.240 | 40.9 | 0.997 |
| BSK2-UD-(0.60) | 0.014 | 3.0 | 0.187 | 41.6 | 0.997 |
| BSK2-UD-(1.05) | 0.014 | 2.962 | 0.113 | 35.5 | 0.998 |
| BSK2-UD-(1.63) | 0.013 | 3.0 | 0.206 | 40.5 | 0.998 |
| Average SWCC | 0.015 | 3.045 | 0.134 | 40.8 | - |

Uniaxial compression tests were carried out on rock core samples in accordance with ASTM D7012 [51]. Test samples were prepared by cutting the core samples in accordance with the test standard. The results obtained from the tests are given in Table 3.

**Table 3.** Lab test results carried out on rock core samples.

| Borehole No | Rock Type | Depth (m) | Natural Unit Weight (kN/m³) | Uniaxial Compression Strength (MPa) |
|---|---|---|---|---|
| BH3 | Siltstone/Claystone | 15.50 | 22.58 | 4.03 |
| BH2 | Sandstone | 12.50 | 25.40 | 57.97 |
| BH2 | Siltstone/Claystone | 6.50 | 22.73 | 4.88 |

## 5. Instrumental Monitoring

A field instrumental monitoring program was conducted for a period of two years to investigate the seasonal behavior of the representative landslide area. Instrumentation consisted of volumetric water content sensors (EC-5), matric water potential sensors (MPS-6), groundwater level sensors (TD-Diver/BaroDiver), double-spoon tipping bucket rain gauge (ECRN-100) and data loggers (EM-60). Some detailed properties of each instrument are presented in Table 4. These instrumentations were installed in the area according to the

plan in Figures 1c, 2 and 7. The data measurement interval of each instrument was set to 60 min.

**Table 4.** Some detailed properties of each instrument.

| Instrument | Measurements | Measurement Range | Resolution | Source |
|---|---|---|---|---|
| EC-5 soil water content sensors | Volumetric water content | 0–60% | 0.001 $m^3/m^3$ | Meter Group Inc. Pullman, WA, USA |
| MPS-6 water-potential sensors | Matric suction | 9 kPa–100,000 kPa | 0.1 kPa | Meter Group Inc. Pullman, WA, USA |
| ECRN-100 rain gauge | Rain depth | N/A | 0.2 mm | Meter Group Inc. Pullman, WA, USA |
| TD-Diver pressure sensors | Water pressure + Atmospheric pressure | 0–50 m $H_2O$ | 1 cm $H_2O$ | Van Essen Inst. Delft, The Netherlands |
| Baro-Diver atmospheric pressure sensor | Atmospheric pressure | 0–1.5 m $H_2O$ | 0.1 cm $H_2O$ | Van Essen Inst. Delft, The Netherlands |
| EM-60 data logger | Input port: six, each supporting meter analog, digital, or pulse sensors | 1 min. to 12 h. | N/A | Meter Group Inc. Pullman, WA, USA |

Two volumetric water content sensors and three matric suction sensors were installed behind the main scarp at depths of 1.10 m, 1.63 m, and 0.58 m, 1.10 m and 1.63 m, respectively, as shown in Figure 7a. Five separate boreholes of 100 mm diameter were drilled using a hand auger to install the sensors. The boreholes were drilled to depths of sensor installation. After the sensors were installed at the bottom of the boreholes, the boreholes were backfilled with the in-situ cut soil, and sensor cables were then connected to the data logger for taking values.

Rainfall and its intensity were measured by a double-spoon tipping bucket rain gauge. The rain gauge was mounted by its bracket to an installation pole at 1.80 m height (Figure 7b). A datalogger was also placed onto the pole, and a sensor cable was connected to the datalogger for data processing.

In order to monitor changes in groundwater level, TD-Diver and Baro-Diver submersible sensors were installed in piezometers in the boreholes of BH1 and BH3. The Divers measure the equivalent hydrostatic pressure of the water above the sensor to calculate the total water depth and record them in its internal memory. The absolute pressure is measured by the Divers. This indicates that in addition to measuring water pressure, the pressure sensor also monitors atmospheric pressure acting on the water's surface. The Baro-Diver only measures the atmospheric pressure, and the TD-Diver measures both the atmospheric pressure and the pressure exerted by the water column above the Diver. In this study, one Baro-Diver was installed above the water level in BH3 to measure the atmospheric pressure of the landslide area. The TD-Divers were installed below the water levels in BH1 and BH3. The Divers were read using a laptop and USB Reading Unit in the field (Figure 7c).

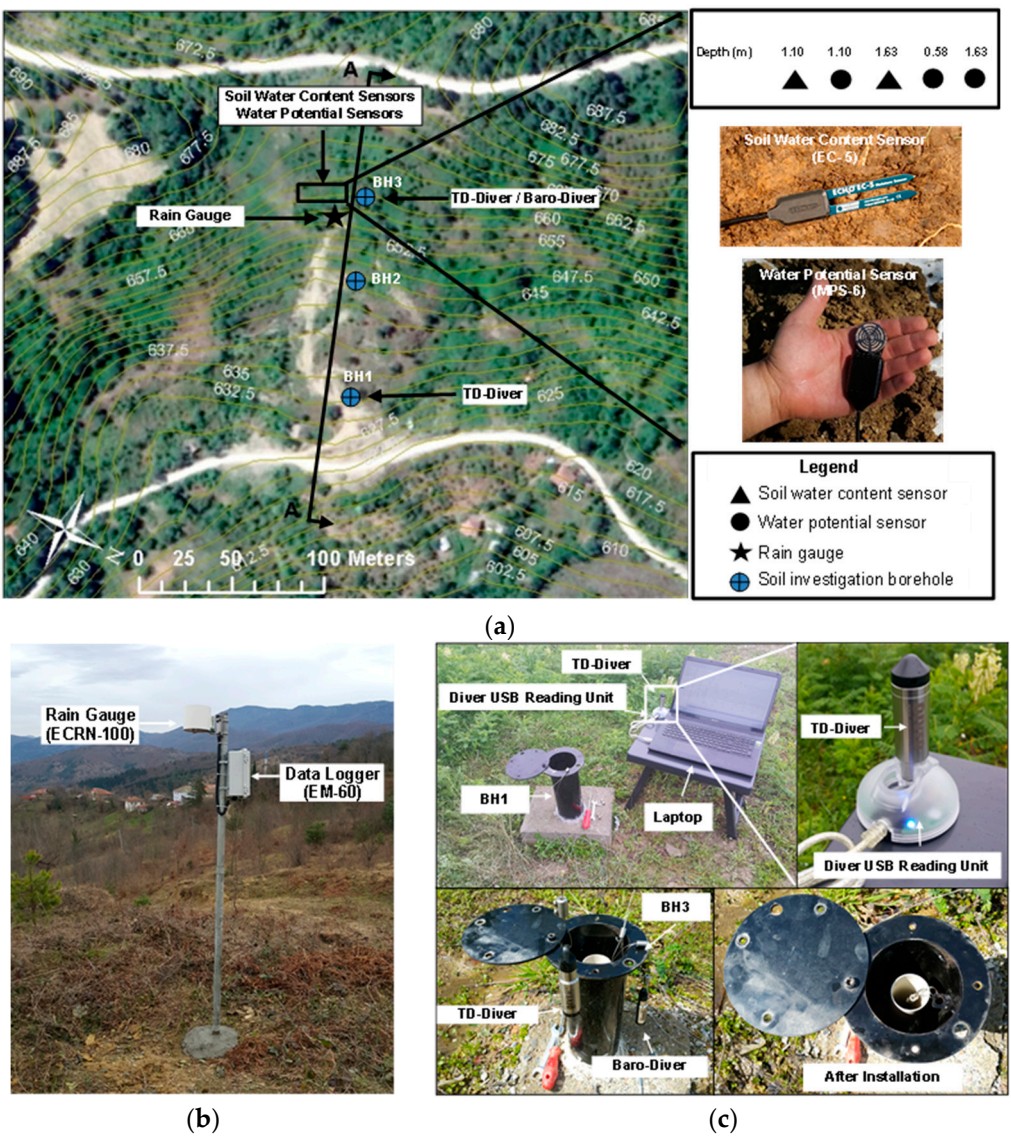

**Figure 7.** Location of boreholes and installed sensors. (**a**) Location map, EC-5 and MPS-6, (**b**) ECRN-100 rain gauge, (**c**) Diver sensors.

### 5.1. Monitoring of Volumetric Water Content and Matric Suction

As shown in Figure 7a, three matric suction sensors and two volumetric water content sensors were installed behind the main scarp at depths of 0.58 m, 1.10 m, 1.63 m and 1.10 m, 1.63 m, respectively. The variation in matric suction at 0.58 m depth is given in Figure 8. Matric suction generally varied between 8.5 kPa and 61.6 kPa during the two-year observation period. Although a matric suction of 208.7 kPa was initially measured, hydraulic equilibrium was achieved in a short time, and matric suction reached around 60 kPa. During the two-year monitoring period, the lowest matric suctions were measured from mid-January to the end of July at a 0.58 m depth. The highest matric suctions were measured from mid-October to the end of December. From the end of July to the middle of October, increasing and decreasing trends were observed in matric suctions.

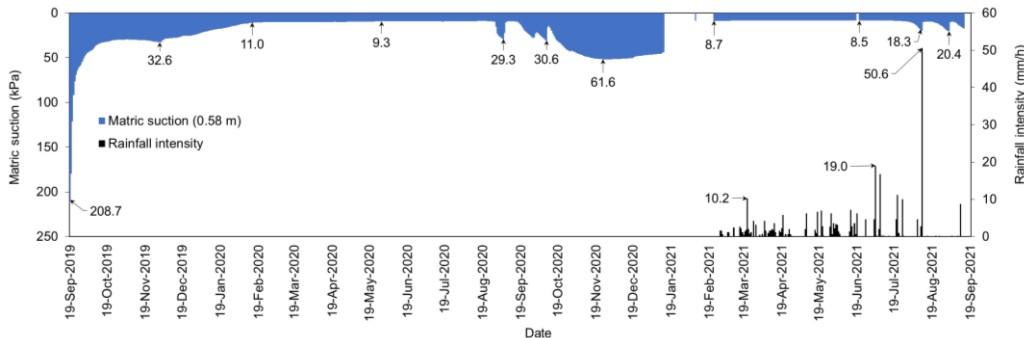

**Figure 8.** Variation in matric suction at 0.58 m depth.

The responses of volumetric water content and the matric suction at a depth of 1.10 m are given in Figures 9 and 10. Initially, the volumetric water content was 14%, and the matric suction was 154 kPa. The volumetric water content gradually increased to 17%, and the matric suction decreased to 60 kPa on 1 February 2020. The reason for the rapid decrease in matric suction equilibrium establishment of the sensors with the surrounding soil, according to the second law of thermodynamics. Between 1 February 2020 and 14 February 2020, the matric suction decreased from 60 kPa to 10 kPa, and the volumetric water content increased from 17% to 45.8%. Between 14 February 2020 and 25 May 2020, the volumetric water content was around 47%, and the matric suction was around 9 kPa. The volumetric water content decreased step by step after 25 May 2020 and reached 15% on 5 September 2020. On the other hand, there was no significant variation in the matric suctions, contrary to the expectation of an increase. A similar behavior was also observed in cases where the volumetric water content decreased step by step after 7 May 2021. This phenomenon may be due to a change in the shape of the SWCC because of crack formation in the soil. Cracks develop in soil due to drying, and this affects the mechanical and hydraulic properties of soils [54,55]. Various studies, such as References [56–60], have reported that the formation of cracks in clayey soils causes significant changes in the soil water characteristic curve. For an illustration, Zornberg et al. [59] observed that crack development increased the slope of the soil water characteristic curve at the transition zone. This results in higher volumetric water content changes but more limited matric suction changes beyond the air-entry value. According to Li et al. [57], the soil cracks deform during the drying and wetting stages, and the changes in crack volume and suction have a direct impact on the soil water characteristic curve of cracked soil. It is possible to explain the differences in the matric suction and volumetric water content values in Figures 9 and 10 based on this information in the literature.

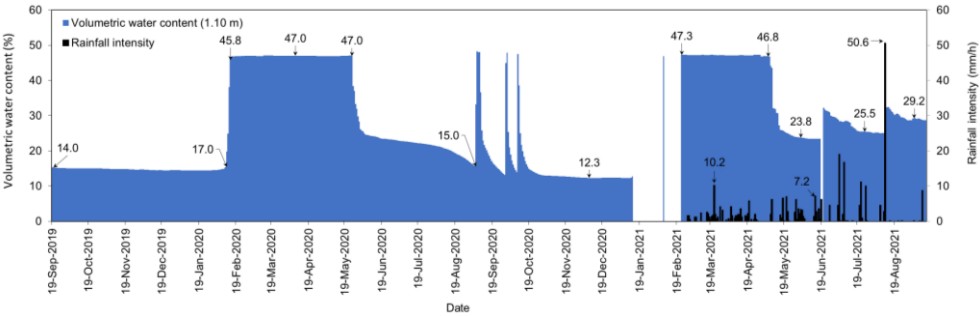

**Figure 9.** Variation in volumetric water content at 1.10 m depth.

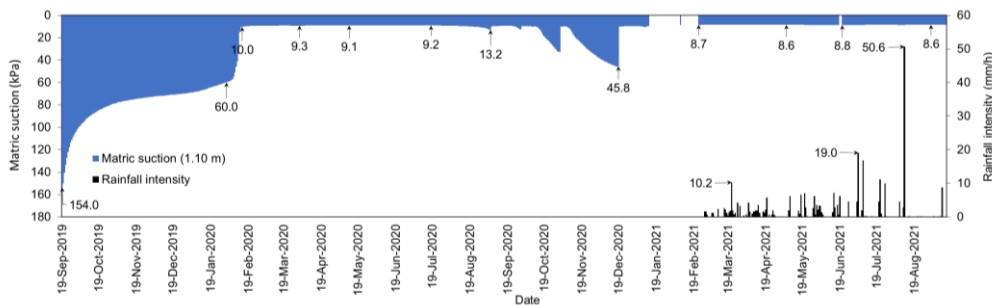

**Figure 10.** Variation in matric suction at 1.10 m depth.

In the wetting stage, due to the insignificant number of cracks or closed cracks, the water infiltrating into the hillslope increased the water content of the soil by entering into the fine interparticle pores where the matric suctions are effective and decreased the matric suction almost simultaneously. In the drying phase, due to the crack development, a significant volume of water first drained from these cracks, and the water content of the soil was significantly reduced. Meanwhile, in fine pores where matric suction is effective since the water content was still preserved or changed slightly, matric suction change became slightly.

As observed in Figures 9 and 10, the start of rainfall measurement on 23 February 2021 in the landslide area enabled the comparison of the volumetric water content matric suction with rainfall after this date. Between 23 February 2021 and 7 May 2021, a total of 250 mm of rain fell on the hillslope, with a daily maximum rainfall of 28 mm and an hourly maximum rainfall of 10.2 mm. In this period, volumetric water content was around 47%, and the matric suction was around 8.6 kPa. Between 7 May 2021 and 14 September 2021, a total of 463 mm of rain fell in the study area, with a daily maximum rainfall of 96.8 mm and an hourly maximum rainfall of 50.6 mm. After May 2021, heavier and less frequent rainfall was observed. Prolongation of the rainfall intervals reduced the volumetric water content by allowing the soil to dry.

The variations in the volumetric water content and matric suction at a 1.63 m depth are presented in Figures 11 and 12. It can be remarked that at this depth, the volumetric water content and the matric suction did not change remarkably and were not affected extensively by seasonal climatic conditions. During the observation period, the volumetric water content was 42–46%, and the matric suction was measured between 21.5 and 10.0 kPa, at this depth.

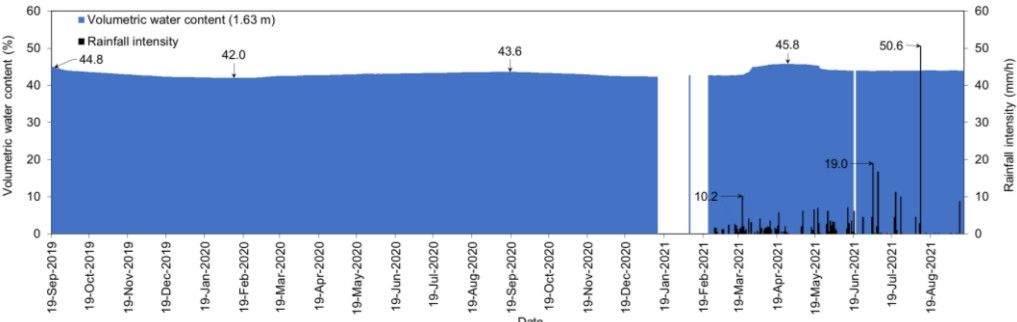

**Figure 11.** Variation in volumetric water content at 1.63 m depth.

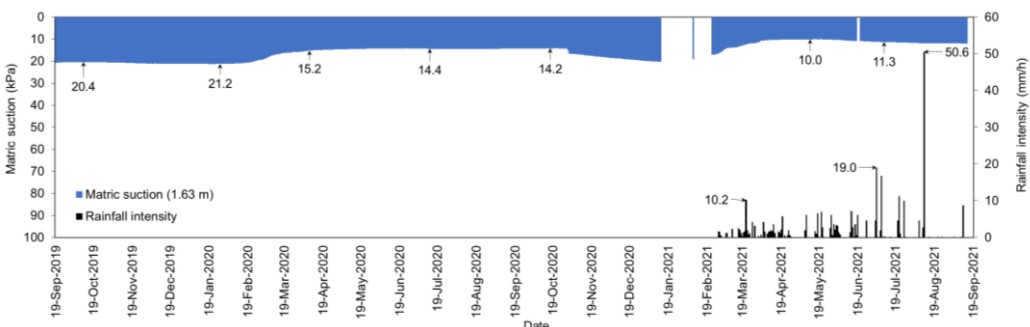

**Figure 12.** Variation in matric suction at 1.63 m depth.

*5.2. Monitoring of Groundwater Level*

There are fluctuations in groundwater levels depending on seasonal climatic conditions. While the groundwater level rises in rainy periods, it decreases in hot and dry periods due to less recharge. Fluctuations in the groundwater level can affect the positive pore water pressure distribution and the negative suction profile in the vadose region, thus affecting the stability. While investigating the hydro–mechanical behavior of landslides, it is of great importance to examine the changes in the groundwater level.

Figure 13 compares the groundwater levels in BH3 and BH1 with hourly rainfall intensity on various dates. Groundwater levels were monitored in BH3 between February 2021 and September 2021. Due to the collapse of the BH1, groundwater levels could only be measured in BH1 between February 2021 and June 2021. As shown in Figure 13, the groundwater level in BH3 was located at a 2.21–3.49 m depth below the ground surface. In BH1, it varied between 6.01 and 8.72 m. The reason why the groundwater level was measured deeper in BH1 can be due to BH1 being located at the foot of the landslide. It is possible that the groundwater level was measured at a deeper level due to the accumulation of soil at the foot. The highest groundwater levels in the BH1 and BH3 were 6.01 and 2.21 m, respectively, on April 2021. The deepest groundwater level was measured as 8.72 m in BH1 on May 2021 and 3.49 m in BH3 on August 2021. As shown in Figure 13, the groundwater levels tended to increase with the low-intensity, long-duration rainfalls observed in February, March and April. More fluctuation in the groundwater level was monitored in BH1. During these months, while the groundwater levels were rising, the water content of the soils in the upper parts was high, and the matric suction was low (see Figures 8–12). The groundwater level started to decrease after April. Decreased groundwater level had almost no effect on soil moisture and matric suction at a 1.63 m depth (see Figures 11–13). At this depth, the soil is in the capillary zone. After the high-intensity short-duration rainfalls in July and August (19.0 mm/h, 50.6 mm/h), no remarkable increase was observed in the groundwater levels, unlike those observed in the low-intensity long-duration rainfalls.

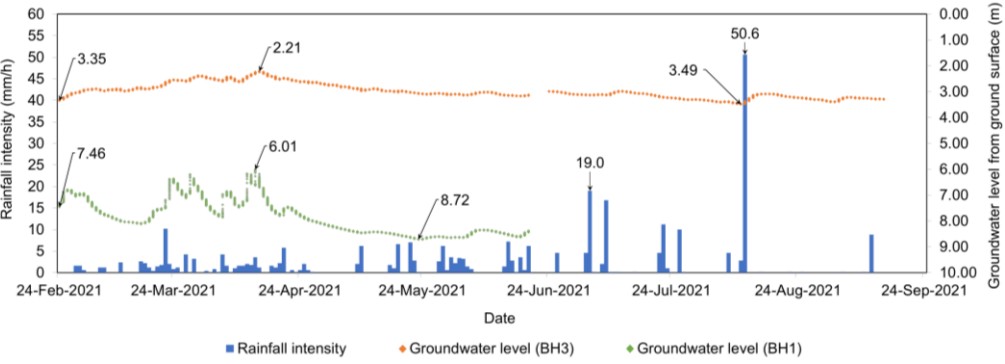

**Figure 13.** Variations in groundwater level with rainfall intensity in BH1 and BH3.

According to the measurements made in the field, it can be remarked that the first five months of the year are wet, and the others are dry for the region. The periodic variation in matric suction and groundwater levels can be considered according to the values given in Table 5. These values were proposed to obtain a general framework for understanding a representative landslide area's hydro–mechanical behavior and were used in the stability analyses. In Table 5, groundwater levels were taken according to measurements in BH3. The reason to consider BH3 for groundwater level is that (i) BH3 is right next to the other monitoring boreholes, and (ii) it is the least affected by landslide activity due to its location. Matric suctions were taken as 0 kPa for the wet period because in-situ volumetric water contents at those depths changed in saturated water content ranges according to lab tests, and matric suction sensors cannot meaningfully measure soil matric suction below 9 kPa, due to their capacities. Matric suctions for the dry period were based on the average measurements recorded during the dry period in the field.

**Table 5.** Periodic matric suctions and groundwater levels used in numerical analyses.

| Depth (m) | Dry Period | | Wet Period | |
| --- | --- | --- | --- | --- |
| | Matric Suction (kPa) | Groundwater Level (m) | Matric Suction (kPa) | Groundwater Level (m) |
| 0.58 | 60 | | 0 | |
| 1.10 | 60 | 3.49 | 0 | 2.21 |
| 1.63 | 15 | | 0 | |

## 6. Numerical Modeling

Using the data obtained as a result of field and laboratory studies, the critical section of the landslide area, shown in Figure 2, was modeled. The idealized ground profile was constituted by considering the geology, survey drilling data and the topography of the study area. The ground consisted of a soil layer with thickness varying up to 6.5 m, overlying the moderately- to highly-weathered levels of the Ulus formation. According to the Unified Soil Classification System (USCS), most soils were classified as CL. The moderately- to highly-weathered levels of the Ulus formation consisted of alternations to claystone, siltstone and sandstone. Details of the numerical modeling and analyses are given below.

### 6.1. Stability Analyses

Stability analyses were performed by using the Bishop limit equilibrium method in the code SLOPE/W [61]. Three different types of stability analyses were carried out: dry period; wet period; and rainfall infiltration analyses. Dry and wet period stability analyses were conducted by assigning average matric suctions and groundwater levels presented in Table 5 as the initial boundary condition to the model. Rainfall infiltration stability analyses were performed by generating new pore-pressure distributions developing in the soil profile after various possible rainfall intensities and durations by means of seepage analyses. The initial pore-pressure distributions before rainfall were formed regarding the average dry- and wet-period suctions and groundwater levels, given in Table 5, in the seepage analyses. The main purpose of these analyses was to observe the effects of the average pore-pressure distributions recorded in dry and wet periods for the two-year monitoring period in the field on the stability of the hillslope and to examine how the stability will be affected by various possible rainfall intensities and durations. Unsaturated shear strength was applied in the slope stability analyses to include the contribution of matric suction, as suggested by Vanapalli et al. [62].

$$\tau = c' + (\sigma - u_a) \tan \phi' + (u_a - u_w) \tan \phi' \left[ \left( \frac{\Theta_w - \Theta_r}{\Theta_s - \Theta_r} \right) \right] \qquad (2)$$

where τ is the shear strength; $c'$ is the effective cohesion; $ø'$ is the effective friction angle; σ is the total normal stress; $u_a$ is the pore air pressure; $(σ − u_a)$ is the net normal stress; $u_w$ is the pore water pressure; $(u_a − u_w)$ is the matric suction; $\Theta_w$ is the volumetric water content; $\Theta_r$ is the residual volumetric water content; $\Theta_s$ is the saturated volumetric water content. In addition to the effective shear strength parameters, SWCC, which shows the relationship between matric suction and water content, should be defined to calculate the shear strength. In this study, the average SWCC in Figure 6 was used as the soil water characteristic curve. The soil properties used in the stability analyses are summarised in Table 6.

**Table 6.** Geotechnical properties used in modeling the soil.

| Parameters | Soil |
| --- | --- |
| Material Model | Mohr-Coulomb |
| $c'$ (kPa) | 8 |
| $ϕ'$ | 21 |
| $γ$ (kN/m$^3$) | 20 |
| SWCC | Figure 6 (average SWCC) |

The strength of the rock mass was determined using the Hoek and Brown [63] failure criterion. The rock masses in the Hisar landslide area comprise intercalations of siltstones and claystones, with interbedded sandstones. The surface condition of their discontinuities is poor. The GSI value was considered to be an average of 30. The disturbance factor D was regarded as 0 since no treatment was applied to the rock mass. The cores obtained from drillings contained approximately 10% sandstone, 45% siltstone and 45% claystone. $m_i$ was calculated as 6.65 by taking the weighted average values introduced by Hoek [64], based on the rock type of the study area. Similarly, uniaxial compressive strength $(σ_{ci})$ was computed as 9.81 MPa by taking the weighted average of laboratory uniaxial compressive test results.

By using the determined Hoek–Brown parameters (GSI = 30, D = 0, $m_i$ = 6.65, $σ_{ci}$ = 9.81 MPa), the normal stress–shear stress relationship of the rock mass was obtained. This normal stress–shear stress function was used as the material model of the rock mass. The geotechnical properties used in modeling the rock mass are summarised in Table 7.

**Table 7.** Geotechnical properties used in modeling the rock.

| Parameters | Rock |
| --- | --- |
| Material Model | Hoek Brown |
| $γ$ (kN/m$^3$) | 23 |
| $ϕ_b$ (°) | 0 |
| GSI | 30 |
| D | 0 |
| $m_i$ | 6.65 |
| $σ_{ci}$ (MPa) | 9.81 |

### 6.2. Seepage Analysis

Seepage analyses were conducted using the finite element method in the software SEEP/W [61]. These analyses aimed to investigate the variations in soil matric suction and positive pore water pressure distribution under different rainfall intensities and durations. Transient flow analyses were employed to determine the changes in matric suction and positive pore water pressure due to rainfall. The initial pore pressure distributions for the transient analyses were constituted based on the in-situ monitoring data, considering the average dry- and wet-period suctions as well as groundwater levels.

Numerical analysis for unsaturated flow through the soil was conducted by assigning two curves, known as the soil water characteristic curve (SWCC) and hydraulic conductivity

function (HCF), into the model that governs flow for unsaturated media. The average SWCC curve presented in Figure 6 was used in the analyses. The Van Genuchten [53] hydraulic conductivity model, given in Equation (3), was implemented to calculate the hydraulic conductivity function.

$$k = k_s \frac{\left\{1 - (\alpha\varphi)^{n-1}\left[1 + (\alpha\varphi)^n\right]^{-m}\right\}^2}{\left[1 + (\alpha\varphi)^n\right]^{\frac{m}{2}}} \tag{3}$$

where k is the unsaturated hydraulic conductivity coefficient; $k_s$ is the saturated hydraulic conductivity coefficient; $\alpha$, n and m are the fitting parameters obtained from SWCC; $\varphi$ is the matric suction.

In the characterization of HCF, the most important parameter required in addition to the soil water characteristic curve is the saturated hydraulic conductivity coefficient of the soil.

The saturated hydraulic conductivity of the soil was measured as an average $1.76 \times 10^{-7}$ m/s in laboratory tests. Regarding studies conducted in the literature, it has been stated that saturated hydraulic conductivities can change with test conditions, test type, and sample size even for identical soils [65,66]. Various studies have reported that saturated hydraulic conductivities measured in field can be $10-10^3$ times higher than those measured in the laboratory [67,68]. In this study, rainfall infiltration analyses were performed for saturated hydraulic conductivities of $1.76 \times 10^{-6}$ m/s, $1.76 \times 10^{-7}$ m/s and $1.76 \times 10^{-8}$ m/s and found to be 10-times smaller and greater than the lab saturated hydraulic conductivity to meet the possible conductivity values for the ground. Thus, the aim was to examine the effect of changes in the saturated hydraulic conductivity on the results and to consider the difference between field and laboratory conditions. Figure 14 shows the variation in hydraulic conductivity function with saturated hydraulic conductivities.

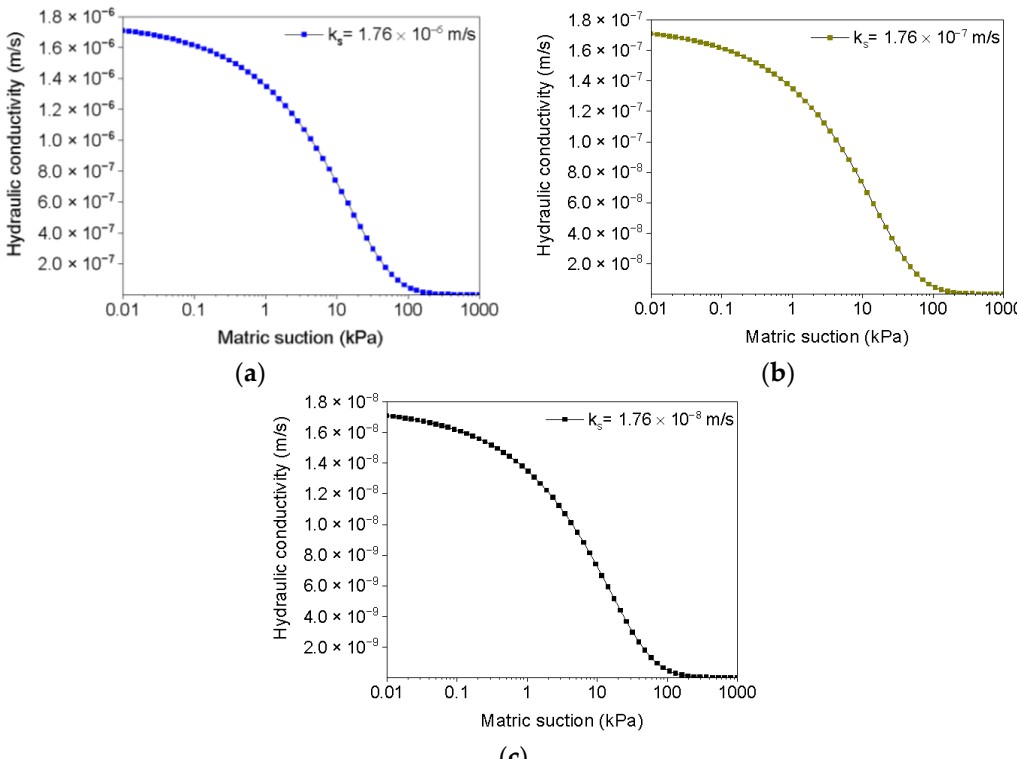

**Figure 14.** Hydraulic conductivity function used in rainfall infiltration analyses. (**a**) $k_s = 1.76 \times 10^{-6}$ m/s, (**b**) $k_s = 1.76 \times 10^{-7}$ m/s, (**c**) $k_s = 1.76 \times 10^{-8}$ m/s.

Rainfall was measured between February 2021 and September 2021 in the Hisar landslide area. Most of the hourly rainfalls were below 20 mm during the monitoring. The maximum hourly rainfall was recorded as 50.6 mm on August 2021. The effects of various rainfall intensities and duration, including the ranges measured in the area, on the pore-pressure distribution and stability of the slope were investigated by seepage and stability analyses. Changes in the pore-pressure distribution were obtained by applying various rainfall intensities, such as 1 mm/h, 2.5 mm/h, 5 mm/h, 10 mm/h, 20 mm/h, 40 mm/h, 80 mm/h and 160 mm/h, for various durations in the transient flow analyses. The pore pressures obtained from the transient analyses were considered as the initial boundary conditions in the stability analyses, and the variations in the factor of safety (FoS) were obtained.

## 7. Results and Discussion

### 7.1. Dry Period and Wet Period Stability Analyses

Pore-pressure distributions and critical factor of safety for stability at dry and wet periods are presented in Figure 15a,b. FoS was calculated as 1.74 for the dry period and 1.32 for the wet period. The analyses are important to reveal how stability is affected by changes in the soil matric suction profile and positive pore water pressure, depending on seasonal conditions in the landslide region.

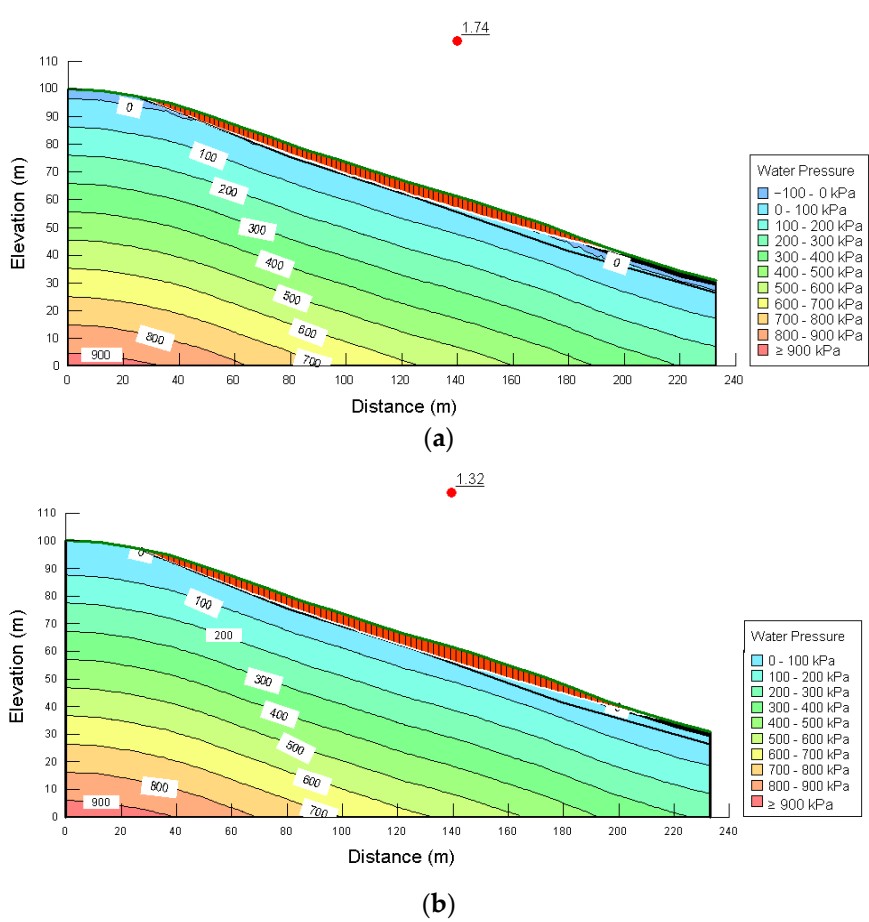

**Figure 15.** Factor of safety and pore pressure distributions. (**a**) Dry period, (**b**) Wet period.

A factor of safety below 1.50 can create a critical situation in terms of stability. The fact that the obtained factor of safety was 1.32 in the analysis considering the wet period conditions shows that slope safety can be critical in a region similar to the examined section and in the case of rainfall close to the critical threshold, the factor of safety can decrease to 1.00. Wet period conditions create a critical situation here very close to pre-failure.

Seasonal variations in matric suction and groundwater level play an important role in slope instability. Suradi et al. [8] highlighted the significance of in-situ conditions at the time before a landslide is triggered. According to Suradi et al. [8], when the degree of saturation in a slope is relatively high, indicating a low initial in-situ suction, it becomes more vulnerable to slope failure triggered by rainfall. Cascini et al. [69] discussed the relationships among rainfall conditions, in-situ soil suction and induced slope instability. The authors proposed four periods in a year for suctions based on in-situ soil suction measurement in the pyroclastic deposits of the Campania region (southern Italy): low (<10 kPa) in January–May; very high (>30 kPa) in June–August; high (20–30 kPa) suction in September–October; and medium (10–20 kPa) in November–December. They showed that slope instability and triggering strictly depend on these suction periods and rainfall patterns. Similar results were confirmed by us for this particular landslide area.

### 7.2. Rainfall Infiltration Stability Analyses

In order to obtain possible stability conditions in the field, considering various possible rainfall intensities (1 mm/h, 2.5 mm/h, 5 mm/h, 10 mm/h, 20 mm/h, 40 mm/h, 80 mm/h, 160 mm/h), durations and hydraulic conductivity variations ($1.76 \times 10^{-6}$ m/s, $1.76 \times 10^{-7}$ m/s, $1.76 \times 10^{-8}$ m/s) for the field, many numerical analyses were made and changes in pore pressure distribution were obtained. Stability analyses were then conducted with the calculated pore pressures and variations in the factor of safety were investigated.

Pore pressure distributions and changes in the factor of safety after 2 h and 0.3833 h with a rainfall intensity of 20 mm/h for dry- and wet-period conditions are presented as an example for a saturated hydraulic conductivity of $1.76 \times 10^{-6}$ m/s in Figures 16 and 17. As observed in Figure 16a, at the beginning of the dry period, the groundwater level was at a depth of 3.49 m from the ground surface. Negative pore water pressures had a minimum value of −60 kPa. According to the analysis, it can be observed that there were some negative pore water pressures (matric suctions) in certain parts of the critical slip surface of the slope for the initial dry period conditions. In this case, the factor of safety was calculated as 1.74. As a result of the rainfall lasting for 2 h, it is observed that the groundwater level rose to the ground surface due to inflow into the ground and the matric suctions disappeared completely (Figure 16b). The FoS decreased to 1.02 as the groundwater level rose to the ground surface.

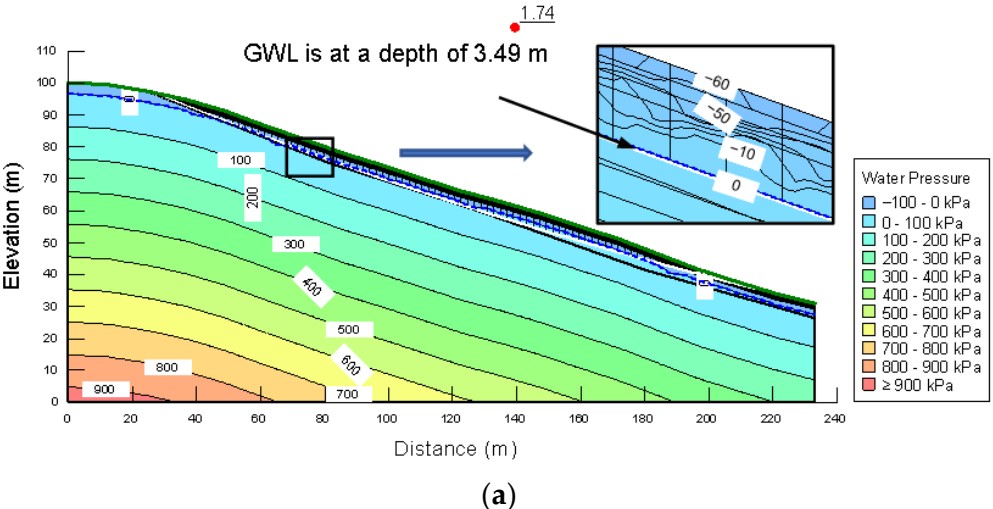

(a)

**Figure 16.** *Cont*.

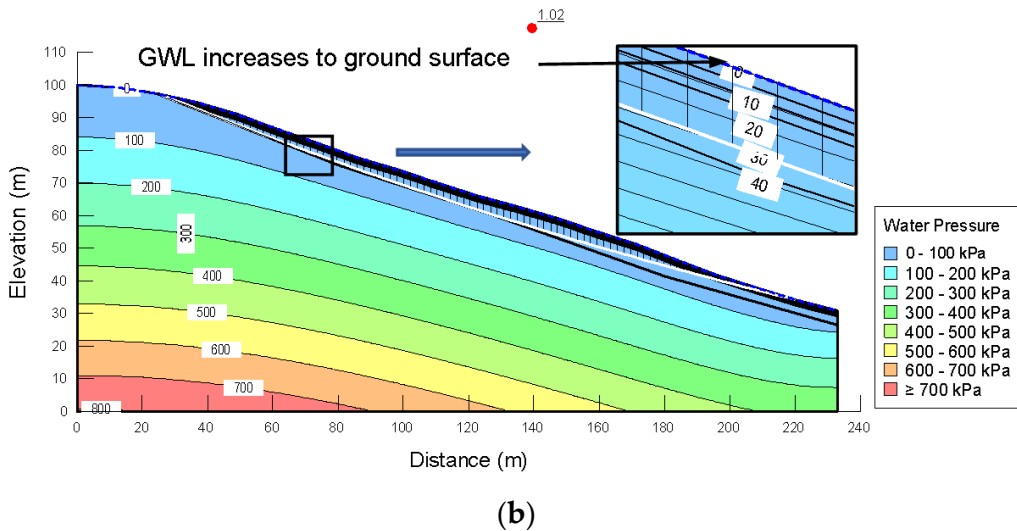

**Figure 16.** Factor of safety and pore pressure distributions. (**a**) Dry period initial conditions, (**b**) the situation that develops after 20 mm/h rainfall intensity lasting 2 h.

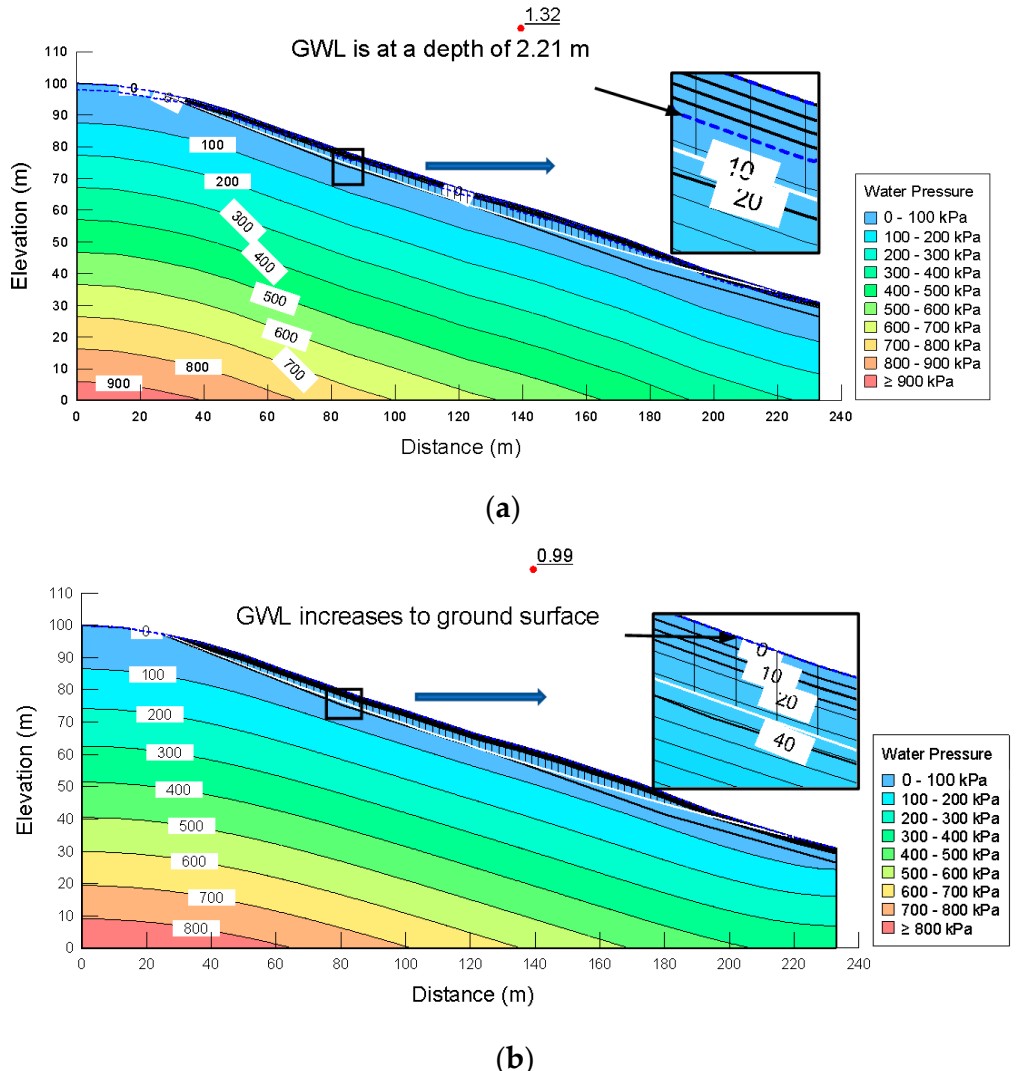

**Figure 17.** Factor of safety and pore pressure distributions. (**a**) Wet period initial conditions, (**b**) the situation that develops after 20 mm/h rainfall intensity lasting 0.3833 h.

On the other hand, as observed in Figure 17a, the groundwater level was at a depth of 2.21 m at initial wet-period conditions. There were higher positive pore water pressures on the critical slip surface. The effect of matric suctions was not observed. As a result of rainfall infiltration performed at the same rainfall intensity (20 mm/h), the factor of safety decreased below the limit equilibrium state (FS = 0.99) after a shorter duration (0.3833 h) (Figure 17b). As rainfall infiltrated the unsaturated soil, it caused an increase in water content and soil saturation, while leading to a decrease in matric suction. Instability can occur when the matric suctions at the critical slip surface of the slope decrease or disappear or when positive pore water pressures develop [70,71]. Results obtained herein are in agreement with studies such as References [6,72,73] that failure is based on the development of positive pore pressure.

Variation in the FoS for various probable rainfall intensities and saturated hydraulic conductivities was examined separately according to the dry and wet periods (Figures 18 and 19). As observed in the figures, the FoS decreased as the rainfall continued and eventually reached its lowest value. For a specific rainfall intensity, higher hydraulic conductivity enabled the lowest factor of safety to be achieved in a shorter duration, as compared to the lower hydraulic conductivity. From the analyses made for dry period conditions, it is observed that the variations in the factor of safety with rainfall intensities are quite close to each other in the case where the saturated hydraulic conductivity is the lowest (Figure 18a). As the saturated hydraulic conductivity increased, scatterings in the factor of safety curves increased, depending on the rainfall intensities (Figure 18b,c). On the other hand, as shown in the wet period analyses, all the factor of safety curves overlapped for the same hydraulic conductivity values, so the changes in the factor of safety with duration were not affected by the rainfall intensities for a specific conductivity (Figure 19a–c). It is possible to explain these changes in the factor of safety with the relationship among initial conditions–drainage–infiltration. In the dry-period initial conditions, the soil was unsaturated above the groundwater level and had matric suctions reaching 60 kPa. Since the soil was not saturated, the hydraulic conductivities varied with the suction. In this case, according to the HCF defined for the saturated hydraulic conductivity value of $1.76 \times 10^{-8}$ m/s (Figure 18a), it was relatively difficult for the rain to infiltrate the ground. Changes in pore water pressure took a long time, accordingly. There was no remarkable increase in the amount of water infiltrating the ground with the variation in rainfall intensity. Excess water that cannot infiltrate is drained by passing into runoff. Therefore, in the case of Figure 18a, where the hydraulic conductivity was lower, the factor of safety varied with a time overlap, and it took a long time to reach the lowest safety factor due to the slow progress of infiltration. For the other two cases with higher hydraulic conductivity coefficients (Figure 18b,c), infiltration occurred more easily and in shorter durations. Changes in pore water pressure were faster than those of lower hydraulic conductivity. As the rainfall intensity increased, the factor of safety curves converged. For wet-period initial conditions, the effects of matric suctions were not observed because the soil was saturated. Since there was no variation between hydraulic conductivity and matric suction, all curves overlapped and were not affected by rainfall intensities. The pore water pressure and, thus, the stability depended on the saturated hydraulic conductivities. When Figures 18 and 19 are compared, it is observed that the wet-period initial conditions reduced the factor of safety below the limit-equilibrium condition more quickly for the same rainfall intensity and saturated permeability values compared to dry-period initial conditions, thus creating a more critical situation before failure.

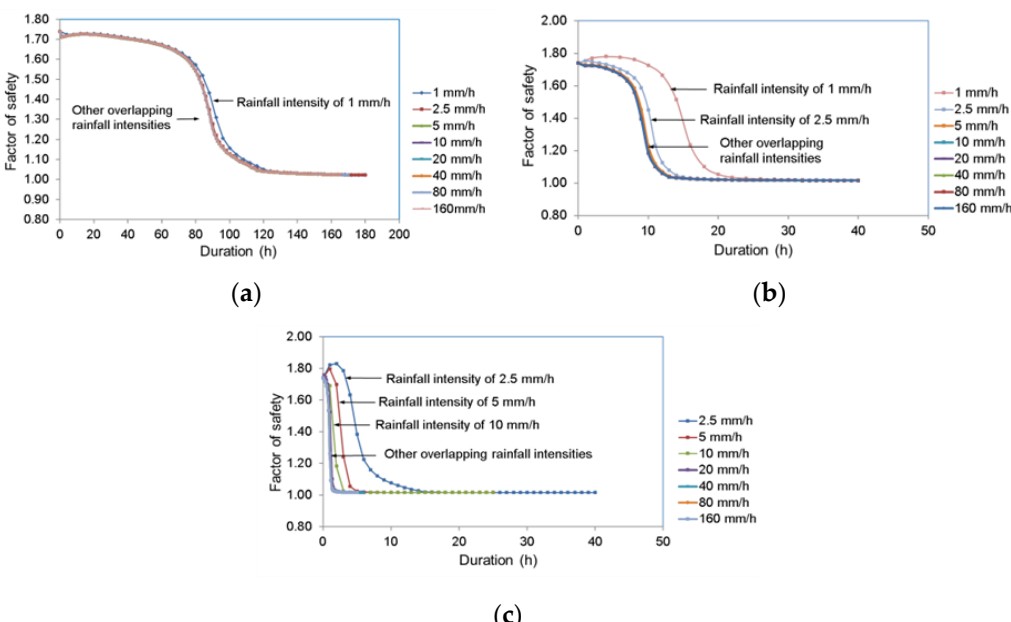

**Figure 18.** Variations in the factor of safety for dry-period initial conditions. (**a**) $k_s = 1.76 \times 10^{-8}$ m/s, (**b**) $k_s = 1.76 \times 10^{-7}$ m/s, (**c**) $k_s = 1.76 \times 10^{-6}$ m/s.

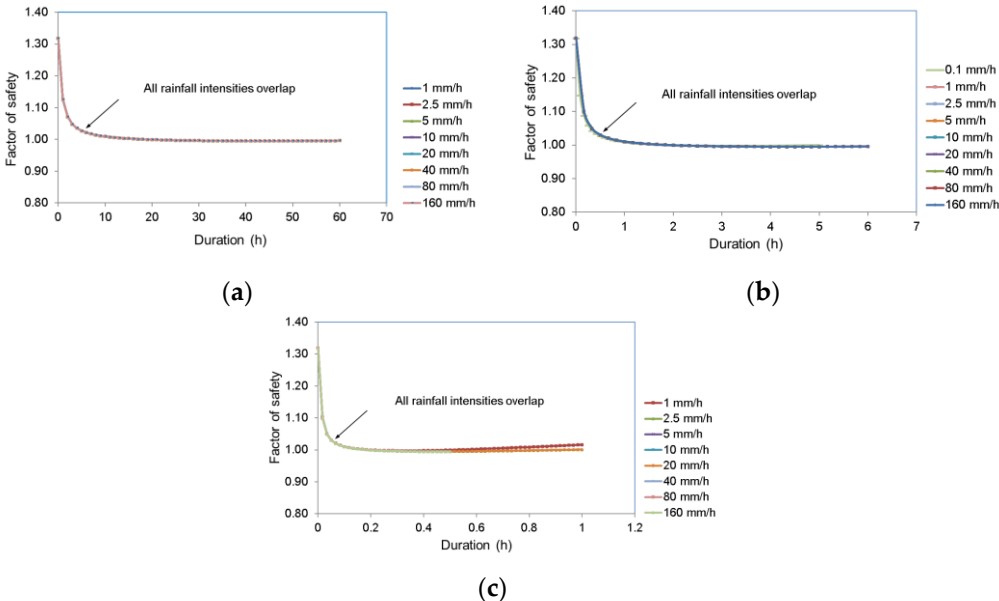

**Figure 19.** Variations in the factor of safety over time for wet-period initial conditions. (**a**) $k_s = 1.76 \times 10^{-8}$ m/s, (**b**) $k_s = 1.76 \times 10^{-7}$ m/s, (**c**) $k_s = 1.76 \times 10^{-6}$ m/s.

In this study, the relationship between the rainfall intensity and duration that can cause landslides in the region was established using the graphs in Figures 18 and 19. Numerous numerical analyses were carried out considering the in-situ measurement parameters. From these graphs, rainfall intensity–duration thresholds were obtained by determining the rainfall intensities and durations which create the factor of safety, 1.50 and 1.02 (the lowest factor of safety reached in the numerical analyses) for the dry period conditions and 1.00 (limit equilibrium state) for the wet period conditions. The rainfall intensity–duration thresholds calculated for the dry and wet periods are presented in Figures 20 and 21, respectively.

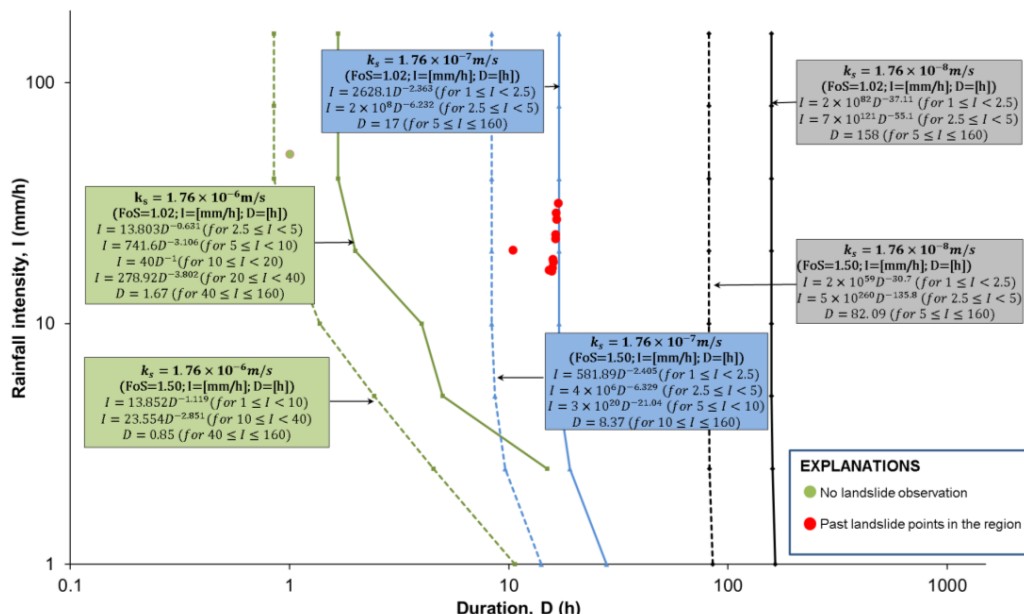

**Figure 20.** Rainfall intensity–duration thresholds for dry period initial conditions.

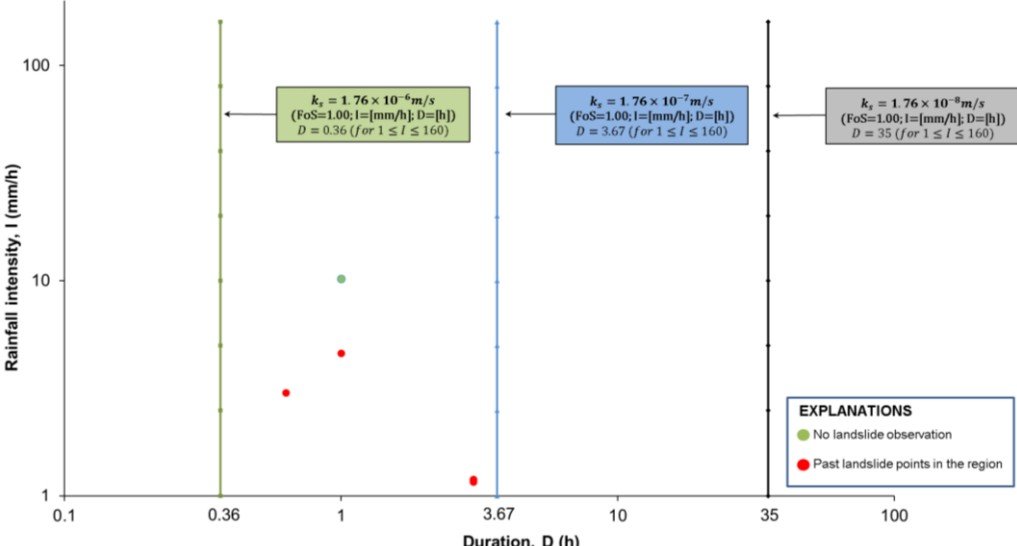

**Figure 21.** Rainfall intensity–duration thresholds for wet period initial conditions.

Intensity–duration (I-D) thresholds are frequently used in the literature. Inspection of the literature reveals that empirical I-D thresholds have a general power function as $I = \alpha \times D^{\beta}$ (where I is the rainfall intensity; D is the rainfall duration; $\alpha$ and $\beta$ are parameters). The suggested I-D thresholds encompassed a significant variety of rainfall durations and intensities. However, the majority of the thresholds included durations ranging from 1 to 100 h and intensities ranging from 1 to 200 mm/h. The value of $\beta$ ranged from $-0.19$ to $-2.00$, and $\alpha$ ranged from 4.00 to 176.40, as reported in the literature [14]. For the landslide area, I-D thresholds were represented by power functions, with $\beta$ in the range of $-0.631$ to $-135.8$ and $\alpha$ in the range of 13.803 to $5 \times 10^{260}$, based on numerical consideration (See Figures 20 and 21). Further, in some ranges of rainfall intensity, particular constant rainfall durations were obtained. The empirical I-D thresholds were defined as more limited ranges of $\beta$ and $\alpha$ when compared to the numerical I-D thresholds proposed in this study. These differences mostly arise from the initial conditions and saturated/unsaturated hydro–mechanical properties of the soil, which are not considered in the empirical I-D thresholds but are considered in numerical analysis methods.

In Figures 20 and 21, it can be clearly observed how the hydraulic conductivity, rainfall intensity and duration affect the I-D thresholds. Decreases in the saturated hydraulic conductivity shifted the rainfall intensity–duration thresholds to the right for both initial conditions. It is also observed that a longer duration of rainfall with the same rainfall intensity was required to trigger the landslide, as the saturated hydraulic conductivity of soil decreased. Rainfall thresholds were obtained as approximately vertical lines for the saturated hydraulic conductivity of $1.76 \times 10^{-8}$ m/s in the dry-period initial conditions and as vertical lines for all saturated hydraulic conductivities in the wet-period initial conditions (see Figures 20 and 21). This is due to the overlapping of the factor of safety curves given in Figures 18 and 19 for the mentioned rainfall intensities and hydraulic conductivities.

The maximum rainfall intensity–duration data points recorded in the dry and wet periods during the observation period in the study area (indicated with green points) and the rainfall intensity–duration data points of landslides that occurred in the past in the region (indicated with red points) were added to the Figures 20 and 21 to verify the established threshold model. The maximum rainfall intensities measured in the dry and wet periods in the study area were 50.6 mm/h and 10.2 mm/h, respectively. After the mentioned rainfall intensities, no total failure was observed in the field. The maximum rainfall intensity–duration data point recorded during the dry period was just to the right of the rainfall threshold obtained for the saturated hydraulic conductivity value of $1.76 \times 10^{-6}$ m/s, with a factor of safety of 1.50 (see Figure 20). The maximum rainfall intensity–duration data point measured in the wet period was between the rainfall thresholds obtained for saturated hydraulic conductivities of $1.76 \times 10^{-6}$ m/s and $1.76 \times 10^{-7}$ m/s and closer to the rainfall threshold with higher permeability (see Figure 21). I-D points of past landslides observed in the region (red points) were also located between the mentioned rainfall thresholds. Except for only one landslide in the wet period, all I-D points of past landslides were to the right of the maximum rainfall intensity–duration data points. An exception can be caused by the differences in saturated hydraulic conductivity at that landslide location. As observed in Figures 20 and 21, it seems highly probable that the permeability in the field was between $1.76 \times 10^{-6}$ m/s and $1.76 \times 10^{-7}$ m/s. In this case, landslide-risk situations are reached if rainfall over 10 mm/h for dry periods lasts between 0.85 h and 8.37 h for an FoS = 1.5 and 1.67 h–17 h for an FoS = 1.0 depending on permeability. On the other hand, at rainfall intensities less than 10 mm/h, longer durations are needed for a landslide to occur. It is observed that landslide-risk situations are reached if the rainfall over 1 mm/h for wet periods lasts between 0.36 h and 3.67 h depending on the saturated permeability.

Rainfall-induced landslides quite frequently occur in Bartin, resulting in loss of life and significant disruption. The suggested I-D thresholds in this study for initiating landslides may contribute to mitigating landslide risks. In cases where local or regional thresholds have developed, these thresholds can be utilized to forecast the potential slope failures within an area, given the rainfall measurements or quantitative weather forecasts. Landslide early warning systems, relying on empirical rainfall thresholds and rainfall measurements or weather forecasts, have been established in various places such as the San Francisco Bay region [74], Hong Kong [75], Rio de Janeiro [76], the Piedmont region [11]. A similar warning system can be established for the study area using the I-D thresholds in this work, reducing the uncertainties not considered in empirical I-D thresholds.

## 8. Conclusions

A representative landslide area was chosen in Bartin, one of the cities where rainfall-induced landslides are most common in the Western Black Sea Region of Türkiye. A detailed field monitoring program was conducted to clarify the triggering mechanisms of rainfall-induced landslides. The instrumentation covered the measurements of site suction, volumetric water content, groundwater level, and rainfall amount over a period of two years. Various stability analyses were performed regarding both pore presures after transient flow infiltration and site-measured suctions. Rainfall thresholds for landslides

were obtained by considering the site-specific seasonal suctions, groundwater levels, and saturated/unsaturated hydro–mechanical properties. The achievements obtained from this study, which were planned to illuminate the hydro–mechanical behavior of rainfall-induced landslides in the region and to contribute to landslide early warning systems in this sense, are given below.

In-situ monitoring of volumetric water content, matric suction and groundwater level depending on seasonal conditions for two years has importance in terms of establishing site-specific initial conditions and stability. According to the measurements in the field, it can be remarked that the first five months of the year are wet, and the other months are dry. It has been observed that wet period conditions create more critical situations for failure compared to dry period conditions, so the potential for landslides is higher for wet periods.

From the stability analyses based on the measured average pore pressures for dry and wet seasons, the factor of safety was calculated as 1.74 and 1.32, respectively. The results are significant as they reveal how stability is affected by the seasonal changes in suction and positive pore-water pressure in the region. It has been observed in the studies that the seasonal initial conditions, rainfall intensity, duration, and range of possible saturated hydraulic conductivities affect the stability in various ways.

Transient flow infiltration analyses showed that the groundwater level could rise to the ground surface after possible rainfalls for the area, and the matric suction could be lost completely. The factor of safety might drop below the limit equilibrium condition (F.S = 1).

Rainfall thresholds for landslides were established as a result of numerical analyses performed, considering measured seasonal suctions, groundwater level changes and saturated/unsaturated hydro–mechanical properties. In the analyses, rainfall thresholds were ascertained by investigating the rainfall intensities and duration, which creates a factor of safety 1.50 and 1.02 (the lowest factor of safety reached in the numerical analyses) for the dry period conditions, and 1.00 (limit equilibrium state) for the wet period. Established models were validated by instrumental monitoring and past landslide data observed in the region.

Based on an interpretation of analyses, the saturated hydraulic conductivities in the studied area are expected to be between $1.76 \times 10^{-6}$ m/s and $1.76 \times 10^{-7}$ m/s. In this case, landslide-risk conditions are reached if rainfall of over 10 mm/h for dry periods lasts between 0.85 h and 8.37 h for an FoS = 1.5 and 1.67 h–17 h for an FoS= 1.0 depending on the saturated hydraulic conductivity. When rainfall intensities are lower than 10 mm/h, longer rainfall durations are required for a landslide to occur. For wet periods, landslide-risk situations are reached when rainfall is over 1 mm/h and lasts between 0.36 h and 3.67 h, depending on the permeability.

**Author Contributions:** Conceptualization, S.A., T.T. and Y.Y.; methodology, S.A., T.T. and Y.Y.; software, S.A.; validation, S.A. and T.T.; formal analysis, S.A.; investigation, S.A.; resources, S.A.; data curation, S.A. and T.T.; writing—original draft preparation, S.A.; writing—review and editing, S.A., T.T. and Y.Y; visualization, S.A.; supervision, T.T. and Y.Y.; project administration, S.A. and Y.Y.; funding acquisition, S.A. and Y.Y. All authors have read and agreed to the published version of the manuscript.

**Funding:** This research was funded by TUBITAK (The Scientific and Technological Research Council of Türkiye) with Project Number 120M436, and by the Gazi University Scientific Research Projects Coordination Unit with Project Number 06/2018-29.

**Institutional Review Board Statement:** Not applicable.

**Informed Consent Statement:** Not applicable.

**Data Availability Statement:** The data supporting the findings of this study are available from the S.A. upon request.

**Conflicts of Interest:** The authors declare no conflict of interest.

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
