# Peer review of "Hydro–Mechanical Behaviour of a Rainfall-Induced Landslide by Instrumental Monitoring: Landslide–Rainfall Threshold of the Western Black Sea Bartin Region of Türkiye"

_applsci, doi:10.3390/app13158703_

Round 1
Reviewer 1 Report
This paper named “Hydro-Mechanical Behaviour of a Rainfall-Induced Landslide 2 by Instrumental Monitoring: Landslide-Rainfall Threshold of 3 the Western Black Sea Bartin Region of Turkiye”, aims to report the triggering mechanisms of rainfall-induced landslides located in Bartin city. On the whole, the manuscript is complete in structure, but suffer weaknesses as follows:
1. In section “Geotechnical Investigations”, what lithology is the bedrock of the landslide? How are the “alternations”? And needs more information about the geotechnical profile.
2. What fitting formula is used in SWCC curve? And the fitting parameters?
3. This manuscript need cross-over study about numerical simulation and field monitoring, not only shows the workload or research independently.
4. Some important literature, DOI: 10.1007/s10346-022-01983-8; DOI: 10.1007/s10346-021-01789-0; DOI: 10.1007/s11069-021-04706-9
good.
Reviewer 2 Report
Authors should discuss the results and how they can be interpreted from the perspective of previous studies and of the working hypotheses. The findings and their implications should be discussed in the broadest context possible. Future research directions may also be highlighted.
The Manuscript “Hydro-Mechanical Behaviour of A Rainfall-Induced Landslide by Instrumental Monitoring: Landslide-Rainfall Threshold of the Western Black Sea Bartin Region of Turkiye” analyses the stability of a landslide prone area located in Bartin (Turkey). First, the results of site measurements and laboratory tests are presented. Then, an FE numerical model is introduced to investigate the stability in a critical section of the selected area, with the properties experimentally derived. Although the approach is accurate from a scientific point of view, it’s difficult to catch the novelty of such research submitted as “Article” and not as “Technical Paper”. The study appears just as an application of well-consolidated models, despite correctly characterised. I suggest changing the article type or clearly indicating what is the proposed novelty. Further suggestions in the following.
1) Lines 43-44. Please mention the limitations of rainfall thresholds providing many false alarms because developed considering only “triggering rainfall” and not the non-triggering ones, despite their high intensity. Refer to: https://doi.org/10.1016/j.enggeo.2020.105965; https://doi.org/10.1016/j.enggeo.2022.106834
2) Lines 51-55. This statement is not true. Note that there are many physically-based model where the unsaturated strength is introduced in classical criteria of soil mechanics. Refer to: https://doi.org/10.1016/j.compgeo.2022.105175; https://doi.org/10.5194/nhess-13-151-2013
3) Lines 81-82, about “transient flow analysis”. A similar approach is used by TRIGRS (Baum, R.L.; Savage, W.Z.; Godt, J.W. TRIGRS—A Fortran Program for Transient Rainfall Infiltration and Grid-Based Regional 537 Slope-Stability Analysis, Version 2.0; U.S. Geological Survey: Reston, VA, USA, 2008). However, it is not mentioned.
4) Line 149. Change “atterberg” in “Atterberg”
5) Lines 164-164. How was the in-situ density determined?
6) Lines 148-149. Can we say “parameter”, being a curve?
7) Lines 239-242. There repeat lines 211-213. Please remove or write them differently.
8) Line 393. Remove the yellow highlight.
9) Lines 402-409. They seem like what was already described in Section 6.1. Try to improve the writing.
10) Figure 14. It is not clear how this variation with the matrix suction was derived. Please clarify. Also, consider increasing the font size of axis labels and ticks.
11) Lines 448-449. Fix the spelling of “ciritical” (448) and “peresented” (449)
12) Please improve the Discussion section, following the Instructions for Authors “Authors should discuss the results and how they can be interpreted from the perspective of previous studies and of the working hypotheses. The findings and their implications should be discussed in the broadest context possible. Future research directions may also be highlighted.”
The Manuscript “Hydro-Mechanical Behaviour of A Rainfall-Induced Landslide by Instrumental Monitoring: Landslide-Rainfall Threshold of the Western Black Sea Bartin Region of Turkiye” analyses the stability of a landslide prone area located in Bartin (Turkey). First, the results of site measurements and laboratory tests are presented. Then, an FE numerical model is introduced to investigate the stability in a critical section of the selected area, with the properties experimentally derived. Although the approach is accurate from a scientific point of view, it’s difficult to catch the novelty of such research submitted as “Article” and not as “Technical Paper”. The study appears just as an application of well-consolidated models, despite correctly characterised. I suggest changing the article type or clearly indicating what is the proposed novelty. Further suggestions in the following.
1) Lines 43-44. Please mention the limitations of rainfall thresholds providing many false alarms because developed considering only “triggering rainfall” and not the non-triggering ones, despite their high intensity. Refer to: https://doi.org/10.1016/j.enggeo.2020.105965; https://doi.org/10.1016/j.enggeo.2022.106834
2) Lines 51-55. This statement is not true. Note that there are many physically-based model where the unsaturated strength is introduced in classical criteria of soil mechanics. Refer to: https://doi.org/10.1016/j.compgeo.2022.105175; https://doi.org/10.5194/nhess-13-151-2013
3) Lines 81-82, about “transient flow analysis”. A similar approach is used by TRIGRS (Baum, R.L.; Savage, W.Z.; Godt, J.W. TRIGRS—A Fortran Program for Transient Rainfall Infiltration and Grid-Based Regional 537 Slope-Stability Analysis, Version 2.0; U.S. Geological Survey: Reston, VA, USA, 2008). However, it is not mentioned.
4) Line 149. Change “atterberg” in “Atterberg”
5) Lines 164-164. How was the in-situ density determined?
6) Lines 148-149. Can we say “parameter”, being a curve?
7) Lines 239-242. There repeat lines 211-213. Please remove or write them differently.
8) Line 393. Remove the yellow highlight.
9) Lines 402-409. They seem like what was already described in Section 6.1. Try to improve the writing.
10) Figure 14. It is not clear how this variation with the matrix suction was derived. Please clarify. Also, consider increasing the font size of axis labels and ticks.
11) Lines 448-449. Fix the spelling of “ciritical” (448) and “peresented” (449)
12) Please improve the Discussion section, following the Instructions for Authors “Authors should discuss the results and how they can be interpreted from the perspective of previous studies and of the working hypotheses. The findings and their implications should be discussed in the broadest context possible. Future research directions may also be highlighted.”
Round 2
Reviewer 1 Report
The authors have addressed my comments. The cross-over study about numerical simulation and field monitoring is beyond the scope of this paper but it would be interesting if the authors in future could make some analysis in connection with some cross-over study where the relation between the different parameters might change completely the way how the system behave.
No